# How Vaccinations Changed the Outcome of COVID-19 Infections in Kidney Transplant Patients: Single-Center Experience

**DOI:** 10.3390/vaccines10070990

**Published:** 2022-06-22

**Authors:** Mariarosaria Campise, Carlo Maria Alfieri, Matteo Benedetti, Alessandro Perna, Roberta Miglio, Paolo Molinari, Angela Cervesato, Silvia Giuliani, Maria Teresa Gandolfo, Anna Regalia, Donata Cresseri, Laura Alagna, Andrea Gori, Giuseppe Castellano

**Affiliations:** 1Department of Nephrology, Dialysis and Kidney Transplantation, Fondazione IRCCS Ca’ Granda Ospedale Maggiore Policlinico, 20122 Milan, Italy; carlo.alfieri1@gmail.com (C.M.A.); matteo.benedetti@unimi.it (M.B.); paolo.molinari1@unimi.it (P.M.); silvia.giuliani@unimi.it (S.G.); mariateresa.gandolfo@policlinico.mi.it (M.T.G.); anna.regalia@policlinico.mi.it (A.R.); donata.cresseri@policlinico.mi.it (D.C.); giuseppe.castellano@unimi.it (G.C.); 2Department of Clinical Sciences and Community Health, University of Milan, 20122 Milan, Italy; 3Division of Nephrology, Department of Translational Medical Sciences, University of Campania “L. Vanvitelli”, 80138 Naples, Italy; alexpe_@hotmail.it (A.P.); angela.cervesato@unicampania.it (A.C.); 4UO Nefrologia e Dialisi ASST Fatebenefratelli-Sacco, 20157 Milan, Italy; roberta.miglio@unimi.it; 5Infectious Diseases Unit, Fondazione IRCCS Ca’ Granda Ospedale Maggiore Policlinico, 20122 Milan, Italy; laura.alagna@policlinico.mi.it (L.A.); andrea.gori@unimi.it (A.G.); 6Department of Pathophysiology and Transplantation, University of Milan, 20122 Milan, Italy

**Keywords:** COVID-19 infection, COVID vaccination, kidney transplant, public health

## Abstract

Kidney transplant recipients are a vulnerable population at risk of a life-threatening COVID-19 infection with an incidence of death four-times higher than in the general population. The availability of mRNA COVID-19 vaccines has dramatically changed the fate of this infection also within this fragile population. Transplanted patients have an impaired immunological response also to mRNA vaccines. In March 2021, however, we started a vaccination campaign. These preliminary results show that both the incidence of death and of hospitalization dropped from 13% to 2.4% and from 45% to 12.5% compared to the previous outbreaks reported by our group. In univariate analysis, two variables were associated with an increased risk of hospitalization: older age and dyspnea (*p* = 0.023, *p* < 0.0001, respectively). In multivariate analysis, dyspnea (*p* < 0.0001) and mycophenolate therapy (*p* = 0.003) were independently associated with the risk of hospitalization. The association was even stronger when the two variables were combined (*p* < 0.0001). Vaccinations did not reduce the incidence of COVID-19 infections among our transplanted patients, but provided certain protection that was associated with a significantly better outcome for this infection.

## 1. Introduction

Kidney transplant recipients are a vulnerable population at risk of a life-threatening COVID-19 infection. During the previous pandemic outbreaks until April 2021, we reported—in our Centre—an incidence of death (13%) almost four-times higher than in the general population, in the presence of a low, but possibly underestimated incidence of infection (5.1%) [1]. This could be due to the reduced immunological response associated with the immunosuppressive therapy responsible for the impaired immune response to messenger RNA (mRNA) vaccines, including the COVID-19 vaccine [2]. However, a third COVID-19 vaccine dose has been reported to improve humoral and cellular SARS-CoV-2 immunity also among renal transplant recipients, changing the outcome of this infection also in immunosuppressed patients [3]. In March 2021, following the Italian Health Minister’s indications, we started an extensive vaccination campaign with three doses of BNT162b2 (Pfizer/BioNTech). Subsequent to vaccine availability, vaccinal hubs were distributed all over Italy. Patients were instructed to book an appointment at the vaccine hub closest to their residence. 

The aim of this second ongoing observational and descriptive study is to evaluate how the vaccinations modified the incidence and gravity of COVID-19 infections among kidney transplant recipients. 

## 2. Materials and Methods

In this preliminary report, 72 adult kidney transplant patients (4.2%) out of 1611 followed at our center (Department of Nephrology, Dialysis and Kidney Transplantation Fondazione IRCCS Ca’Granda Ospedale Maggiore Policlinico, Milan, Italy) presented symptoms compatible with COVID-19 infection in the period dating from 31 October 2021 to 15 January 2022 (third outbreak). All of them were included in this analysis. The diagnostic criteria for a suspicion of COVID-19 infection were the same as previously reported [1]. The certainty of diagnosis was based on the positivity of either the antigenic or molecular nasopharyngeal COVID-19 tests. Since the onset of the pandemic, non-hospitalized patients have been treated with the withdrawal or reduction of the anti-proliferative agent in use and the increase of prednisone dosage. After COVID-19 positivity was ascertained, the patients were evaluated by the hospital infective disease specialist for treatment with Remdesivir (Veklury^®^), or the monoclonal association Casirivimab-Imdevimab (Ronapreve^®^) for the delta variant, or Sotrovimab (Xevudy^®^) for the omicron variant. All procedures performed in this study involving human participants were in accordance with the ethical standards of the institutional and/or national research committee and with the 1964 Declaration of Helsinki and the Declaration of Istanbul and its later amendments or comparable ethical standards.

### Statistical Analysis

In the statistical analysis, after an evaluation of their distribution by the means of the Shapiro–Wilk test, continuous variables are expressed as the mean ± SD or as median values and the interquartile range (25–75%ile). Nominal variables are reported as the number of cases (*n*) and relative percentage. Differences among groups (hospitalized vs. not hospitalized patients) were determined by Student’s *t*-test, non-parametric analyses, or Fisher’s test, where appropriate. A general linear model was used to carry out multivariate analysis. Pearson’s Chi-squared test was used to determine whether there was a statistically significant difference between the expected frequencies and the observed frequencies in one or more categories of a contingency table. Statistical analyses were performed using the software IBM SPSS Statistics Version 25. Significance was set for *p*-values < 0.05.

## 3. Results

In the period from 29 October 2021 to 15 January 2022, 72 patients developed the COVID-19 infection, 41 (57%) of which were male, and 58 (80.5%) had received a deceased donor transplant. At the time of infection, the median age was 52 (43–60) years and the median transplant vintage was 57 (27–159) months. Median serum creatinine was 1.37 (1.0–1.7) mg/dL. The immunosuppressive schedule included: calcineurin inhibitors (CNIs), prednisone, mycophenolate (MMF), and mTOR inhibitors in 93–87–79%, and 5.6%, respectively. Five patients had insulin-dependent diabetes. The COVID-19 antigenic nasopharyngeal swab was positive in 65% of patients and the molecular one in 70%. Fifteen (20%) patients were tested with both. At the time of infection, 43 patients were vaccinated with three doses of Comirnaty (BNT162b2)^®^, 21 with two, and 4 with one dose. Four patients were not vaccinated for personal reasons (Table 1). Thirty patients had the δ variant, and the remaining had the omicron one. Table 2 shows the most frequent symptoms at diagnosis. When comparing symptoms between hospitalized and non-hospitalized patients, dyspnea was by far the most prevalent symptom in the case of hospitalization (Table 3). After the increase of the daily steroid dosage (69% of cases) together with MMF temporary withdrawal in 70% of cases and halving in 5%, 23 patients were also treated with monoclonal antibodies (Ronapreve^®^ (T. Hoffmann-La Roche AG, Basel, Switzerland) or Xevudy^®^ (GlaxoSmithKline Brentford, UK)) upon evaluation by an infectious disease specialist, according to the AIFA recommendation [4,5]. In the present cohort, nine out of the 30 patients with the δ variant needed hospitalization for severe respiratory distress and two of them died (6.6%). These patients were compared to 37 out of 82 patients hospitalized during the first and the second wave, all affected by the same variant. The hospitalization and mortality rates were 45% and 29.3%, respectively, during the first two waves, while 30% and 6.6%, respectively, for the present series. Cumulative mortality: 13.4% versus 2.7%. The main difference between the two groups is that the first group of patients did not receive any vaccination, while the second group of patients was vaccinated with at least one vaccine dose: XY had all received XY doses. The two patients who died from respiratory complications had revascularized coronary artery disease. One also had paroxysmal atrial fibrillation, and the other had a pacemaker. 

Five patients who had the COVID-19 infection in 2020 had a second infection. Two were vaccinated with 3 doses, one with 2 doses, and one with 1 dose. The new infection presented the same symptoms: fever, cough, and taste and smell alteration. Only one patient needed hospitalization during both infections because of respiratory insufficiency. In univariate analysis, two variables were associated with an increased risk of hospitalization: older age and dyspnea (*p* = 0.023, *p* < 0.0001, respectively). Hospitalized patients were significantly older and had a longer, but not significant transplant vintage. In multivariate analysis, dyspnea (*p* < 0.0001) and MMF therapy (*p* = 0.003) were independently associated with the risk of hospitalization. The association was even stronger when the two variables were combined (*p* < 0.0001) (Table 3).

## 4. Discussion

Kidney transplant patients are at risk of developing a severe and life-threatening COVID-19 infection. Their poor humoral immunological response accounted for 13% of deaths among non-vaccinated patients reported in our previous paper [1]. In March 2021, following the availability of mRNA vaccines and their approval by the health authority, we started an extensive vaccination campaign with three doses of BNT162b2 (Pfizer(BioNTech). This is a nucleoside-modified-RNA vaccine that induces strong CD4+ T cells and antibody responses. It shows up to 95% efficacy in preventing COVID infection in the immunocompetent population [6]. The effectiveness of vaccines is lower in transplant recipients, but a third vaccine dose [7] or even a fourth one [8] seems to substantially improve humoral and cellular immunity. Our preliminary report shows two important vaccination effects: the reduction of the mortality rate and of hospitalizations. The mortality rate dropped from 13% to 2.7%. The hospitalization rate was 12.5% compared to 45% during the previous outbreaks [1]. 

The pandemic had a detrimental impact on the Health National System both in terms of expenditure and lack of human resources. Reduction of mortality together with reduction of hospitalization reduced this burden. Populations at risk should be extensively vaccinated with three or four doses in the case of absent protective antibody title after three doses. Unfortunately, there is a recent bulk of evidence about vaccine hesitancy in the general population and also among kidney transplant patients. Reasons include: concerns about the speed of the COVID-19 vaccine development (in particular, at the beginning of the vaccinal campaign), insufficient vaccine tests, and fear that complications associated with the vaccination could harm the transplant [9]. Vaccine coverage among kidney transplanted patients could be increased with a clear presentation of the risk–benefit ratio and more information about vaccines and vaccine efficacy in the same risk-population group.

In this cohort, 30 patients contracted the δ variant. Compared to the previous outbreak (1), in which 11 out of 37 hospitalized patients with the same virus variant died (29.7%), in this cohort, only nine patients required hospitalization, but only two died (2.7%). We acknowledge that the small sample size does not allow any definitive conclusion, but the mortality rate showed a progressive decline across infection waves, reaching values lower than what was reported in the general population [10]. We therefore believe that, as well as being linked to improved standard treatments, this is a vaccine-related benefit. Furthermore, to the best of our knowledge, this is the first paper aimed at identifying the impact that COVID-19 infection had on a population at risk such as kidney transplant patients. 

Considering the most important risk factors for COVID-19-related mortality, the two cohorts of 154 patients differed only in diabetes prevalence (23% vs. 7.3% *p* = 0.007). Hospitalization was confirmed to be more frequent in older patients. Vaccinations and post-infection natural immunity did not prevent a second infection in five patients. However, only one needed hospitalization during both infections. Finally, a favorable infection outcome was demonstrated in 24 patients at higher risk of complications, treated with monoclonal therapy after immunosuppression reduction, which remains of paramount importance while managing the infection [11,12]. Treatments have been proven to be safe so far. The main limitation is that we were not able to measure antibody title, and we cannot therefore say that mortality and hospitalization reduction is also due to the development of a protective antibody title post-vaccination. In addition, we acknowledge that the small sample size and the preliminary analysis are the main limitations of our study. 

Furthermore, the availability of new therapeutic options has definitely influenced our results. These early results suggest that a full course of mRNA COVID vaccination in kidney transplant patients together with the modulation of the immunosuppression and the treatment with antiviral or monoclonal antibodies can reduce the severity of this infection. 

## Figures and Tables

**Table 1 vaccines-10-00990-t001:** Patients’ characteristics and treatment of COVID-19 infection.

Variable	
Number of patients (*n*)	72
Gender *n* (%) (M–F)	41 (57)–31 (43)
Median age years (range)	52 (43–60)
Type of transplant, *n* (%) (deceased donor; living donor)	55 (80); 17 (20)
Median transplant vintage (months)	57 (57–160)
Steroids; CyA-Tac; MMF-MPA; mTor inhibitors *n*	63; 10–62; 57; 5
iRAS therapy *n* (%)	21 (29)
Hypertension *n* (%) Diabetes *n* (%)	56 (78) 5 (7)
Mean s-Creatinine (mg/dL)	1.46 ± 0.52
**COVID infection treatment**	
MMF reduction *n* (%)	51 (71)
Steroid increase *n* (%)	50 (69)
Monoclonal antibodies *n* (%)	24 (33)

M: male; F: female; CyA: Cyclosporin; Tac: Tacrolimus; MMF-MPA: mycophenolate-mycophenolic acid; iRAS: renin angiotensin system inhibitors; s-: serum.

**Table 2 vaccines-10-00990-t002:** COVID-19 most frequent symptoms in the total cohort; comparison of frequencies between hospitalized and non-hospitalized patients and outcome.

Symptom	Total Cohort	Hospitalized *n* = 9	Non-Hospitalized *n* = 63	*p*
Fever *n* (%)	44 (36.1)	8 (89)	36 (57)	0.4
Cough *n* (%)	45 (62.5)	6 (67)	39 (62)	0.41
Dyspnea *n* (%)	10 (13.8)	8 (89)	2 (3.1)	**<0.0001**
Asthenia *n* (%)	21 (29.1)	4 (44)	17 (27)	0.43
Diarrhea *n* (%)	9 (12.5)	1 (11)	8 (13)	0.82
Dysgeusia *n* (%) Anosmia *n* (%)	17 (23.6) 17 (23.6)	3 (17) 4 (22)	14 (22) 13 (30)	0.36 0.13
	**Outcome**			
Death *n* (%)	2 (2.7)	2 (22.2)	0	**0.013**

Bold characters are used for significant results.

**Table 3 vaccines-10-00990-t003:** Multivariate analysis.

Variable	F	*p*
Age	0.26	0.61
Transplant vintage	0.13	0.098
Diabetes	0.19	0.66
**MMF-MPA**	9.81	**0.003**
**Dyspnea**	14	**<0.0001**
**Dyspnea + MMF**	14	**<0.0001**

MMF-MPA: mycophenolate-mycophenolic acid. Bold characters are used for significant results. bold characters are used for the significant variables and *p* values.

## Data Availability

If needed, data are available in anonymous form.

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
