# Peer review of "How Vaccinations Changed the Outcome of COVID-19 Infections in Kidney Transplant Patients: Single-Center Experience"

_vaccines, 2022, doi:10.3390/vaccines10070990_

Round 1
Reviewer 1 Report
Q1. Lines 44-45. How was the immunization campaign organized? The patients were vaccinated during recovery? At discharge? At home? During dyalisis?
Q2. Lines 49-50. "at our centre". Which one? Specify
Q3. Line 76. Add proportion after 58
Q4. Table 3. that p>0.0001 is p<0.0001?
Q5. What are the implication of your study from a public health point of view? What do you suggest to raise vaccine coverage in this population?
Q6. Improve discussion paragraph with other experience on the topic reported in literature.
Q7. Compare your results with the ones reported in studies investigating health people and/or other chronic conditions.
Q7. Specify in methods that the study is a preliminary report.
Author Response
ANSWERS TO REVIEWER
We thank the Reviewer for all his suggestions and we appreciate the insightful comments that have been made. In response, we did our best to improve the manuscript.
Comments and Suggestions for Authors
Q1. Lines 44-45. How was the immunization campaign organized? The patients were vaccinated during recovery? At discharge? At home? During dialysis?
- Answer: we added the explanation required: Subsequent to vaccine availability, vaccinal hubs were distributed all over Italy. Patients were instructed to book an appointment at the vaccine hub closest to their residence.
Q2. Lines 49-50. "at our centre". Which one? Specify
- Answer: we specify as requested: (Department of Nephrology, Dialysis and Kidney Transplantation Fondazione IRCCS Ca’Granda Ospedale Maggiore Policlinico, Milan, Italy).
Q3. Line 76. Add proportion after 58
- Answer: proportion added: (80.5%)
Q4. Table 3. that p>0.0001 is p<0.0001?
- Answer: we are very sorry but it was a typing mistake: it is p<0.0001
Q5. What are the implication of your study from a public health point of view? What do you suggest to raise vaccine coverage in this population?
- Answer: We added a comment in Discussion line 155:…. Vaccine coverage among kidney transplanted patients could be increased with a clear presentation of the risk-benefit ratio, more information about vaccines and vaccine efficacy in the same risk-population group.
Q6. Improve discussion paragraph with other experience on the topic reported in literature.
- Answer: Unfortunately we have not been able to find a similar paper. We added a comment in Discussion line 173:….. to the best of our knowledge this is the first paper aimed at identifying the impact that COVID-19 infection had on a population at risk such as kidney transplant patients.
Q7. Compare your results with the ones reported in studies investigating health people and/or other chronic conditions.
- Answer: We thank the Reviewer for his comment. Transplanted patients are an unique population at risk that it is difficult to compare with others. Reason are that their immunological behavior is different also in comparison with other solid organ transplantation patients. In fact immunosuppressive therapy in this special group of patient is long lasting with minimal modifications since day one after transplant while liver transplant or bone marrow transplant patient can have important minimization or suspension. Immunosuppression confers to transplant patients imbalanced immunological response either to vaccines or to infection itself.
Q8. Specify in methods that the study is a preliminary report.
- Answer: it has been specified that the paper is a preliminary report: Materials and Methods: line 52. In this preliminary report, 72 adult kidney transplant patients (4.2%) out of 1611

Reviewer 2 Report
Authors of the paper present descriptive analysis of a cohort of 72 kidney transplant patients suggesting that "a full course of mRNA COVID vaccination .... can reduce the severity of infection."
This statement is not supported with presented results because the group of 72 patients is heterogenous regarding two variants of the virus which showed very different presentations regarding the severity of the disease and outcome. This is a strong confounding factor which is not included in multivariate analysis and this is the major limitation of the study. The results should be stratified but in that case the number of patients would be very low.
The groups which were compared shopuld be defined in scetion materials and methods.
Regarding statistical analysis authors used ttest which should be used when the assumption of normlaity of distribuition was satisfied. It is hard to believe that that asumption was satisfied in a group of 9 patients. This is to small group for any analysis. Differences between groups were not numerically presented for continous variables in table 2. Percentages are not clearly presented - they should be put in brackets.
Although the idea of the study of clinical importance and scientific interest, presented cohort does not allow any well based conclusion.
Author Response
ANSWERS TO REVIEWER 2
We thank the Reviewer for all his suggestions and we appreciate the insightful comments that have been made. In response, we did our best to improve the manuscript.
Comments and Suggestions for Authors
Authors of the paper present descriptive analysis of a cohort of 72 kidney transplant patients suggesting that "a full course of mRNA COVID vaccination .... can reduce the severity of infection."
This statement is not supported with presented results because the group of 72 patients is heterogenous regarding two variants of the virus which showed very different presentations regarding the severity of the disease and outcome. This is a strong confounding factor which is not included in multivariate analysis and this is the major limitation of the study. The results should be stratified but in that case the number of patients would be very low.
The groups which were compared should be defined in section materials and methods.
- Answer: We thank the Reviewer for this comment. As it is now reported in the text, hospitalized patients were compared to the control group of hospitalized patients during the first and second wave from 1 March to 30 April 2020 (first outbreak) and from 1 September 2020 to 21 April 2021 (second outbreak). In addition in the results we added the following paragraph: ….nine out of the 30 patients with the δ variant needed hospitalization for severe respiratory distress and two of them died (6.6%). These patients were compared to 37 out of 82 patients hospitalized during the first and the second wave all affected by the same variant. Hospitalization and mortality rate were 45% and 29.3% respectively during the first two waves, while are 30% and 6.6% respectively in the present series. Cumulative mortality was13.4% versus 2.7%. The main difference between the two groups is that first group patients did not receive any vaccination, while second group patients were vaccinated with at least one vaccine dose: four patients had received 3 doses and 5 received 2 doses.
Regarding statistical analysis authors used ttest which should be used when the assumption of normality of distribuition was satisfied. It is hard to believe that that assumption was satisfied in a group of 9 patients. This is to small group for any analysis. Differences between groups were not numerically presented for continous variables in table 2. Percentages are not clearly presented - they should be put in brackets.
- Answer: We are sorry but obviously there was a mistake in material and methods section. All the continuous variables were tested for their normality by Shapiro-Wilk. In particular which was the only significant variable, had no normal distribution. This is also the reason why we reported it as median (25-75%ile) and its comparison between hospitalized and not hospitalized patients, has been made using non-parametric analyses. The differences between groups for continuous variables are not presented in table 2 because the purpose of table was to show the differences of symptoms and mortality between the two groups.
Although the idea of the study of clinical importance and scientific interest, presented cohort does not allow any well based conclusion.
- Answer: We tried to reinforce the clinical importance and the scientific interest of our paper adding some comments in the discussion and conclusion sections.

Round 2
Reviewer 1 Report
the authors satisfied my requests. In my opinion the manuscript Is now fit for publication.
Author Response
We than the Reviewer.